# Taurine and Ginsenoside Rf Induce BDNF Expression in SH-SY5Y Cells: A Potential Role of BDNF in Corticosterone-Triggered Cellular Damage

**DOI:** 10.3390/molecules25122819

**Published:** 2020-06-18

**Authors:** Won Jin Lee, Gyeong Hee Lee, Jinwoo Hur, Hyuk Gyoon Lee, Eunsu Kim, Jun Pil Won, Youngjae Cho, Mi-Jung Choi, Han Geuk Seo

**Affiliations:** Department of Food Science and Biotechnology of Animal Resources, College of Sang-Huh Life Science, Konkuk University, 120 Neungdong-ro, Gwangjin-Gu, Seoul 05029, Korea; 486251793@naver.com (W.J.L.); kyung9642@naver.com (G.H.L.); jinwoo910218@naver.com (J.H.); krci-12@daum.net (H.G.L.); np-gennao@hanmail.net (E.K.); ww94ww@naver.com (J.P.W.); moonjae@konkuk.ac.kr (Y.C.); choimj@konkuk.ac.kr (M.-J.C.)

**Keywords:** BDNF, corticosterone, ginsenoside Rf, neuronal damage, SH-SY5Y cells, taurine

## Abstract

This study shows that taurine and ginsenoside Rf act synergistically to increase the expression of brain-derived neurotrophic factor (BDNF) in SH-SY5Y human neuroblastoma cells in a dose- and time-dependent manner. The increase of BDNF mRNA by taurine and ginsenoside Rf was markedly attenuated by inhibitors of extracellular signal-regulated kinase and p38 mitogen-activated protein kinase. In addition, taurine and ginsenoside Rf protected cells from corticosterone-induced BDNF suppression and reduced cell viability and lactate dehydrogenase release. The results from this study showed that combined treatment with both taurine and ginsenoside Rf enhanced BDNF expression and protected cells against corticosterone-induced damage.

## 1. Introduction

The mammalian nervous system contains a high concentration of taurine, a sulfur-containing free amino acid [1,2] that is freely available in neurons because it cannot be incorporated into newly synthesized proteins [3]. Taurine is a crucial modulator of neuronal antioxidation and osmoregulation and is neuroprotective [2], alleviating endoplasmic reticulum stress to mitigate hypoxia- and glutamate-induced cytotoxicity [4,5] and preventing damage induced by γ-irradiation and galactose [3,6]. Taurine protects murine hippocampal HT-22 cells against oxidative stress by upregulating heme oxygenase-1 via extracellular signal-regulated kinase (ERK) and p38 mitogen-activated protein kinase [7]. Furthermore, taurine stimulates the growth of neural progenitor/stem cells and promotes neuronal specification [8,9], demonstrating its targeting of multiple mechanisms.

Neuronal function is also modulated by ginsenoside, a bioactive component of ginseng which has been used as a medicinal herb for thousands of years [10,11]. Among the multiple ginsenosides, Rb1, Rb2, Rg1, and Rh2 exhibit the most potent effects in cellular and animal models of neuronal disorders [12,13,14,15,16,17,18,19]. For example, ginsenoside Rb1 alleviates neurological deficits in animals with ischemic insults and traumatic brain injury [12,13,14] and ginsenoside Rg1 suppresses glial activation to prevent neuronal apoptosis and the development of depression-like behaviors in rats [16,17,18]. Similarly, ginsenosides Rb2 and Rh2 protect against glutamate-induced neuronal cell death and trimethyltin-induced neurotoxicity by suppressing oxidative stress and neuroinflammation, respectively [15,19]. However, the effects of ginsenoside Rf are not known.

Brain-derived neurotrophic factor (BDNF) is pivotal for neuronal differentiation, survival, and function [20]. BDNF overexpression in neural stem cells promotes neuronal survival and functional recovery in an animal model of traumatic brain injury [21,22] and BDNF is linked to diverse neurological disorders, such as mood disorders, schizophrenia, and drug-induced brain injury [23]. An upregulation of BDNF also promotes neuron survival and regeneration following brain injury and is involved in the promotion of neuronal dysfunction [24,25]. The neuromodulatory effects of BDNF can be induced by a variety of compounds, such as dexamethasone, vitamin B12, and diverse natural products [25,26,27,28], including ginsenosides. Ginsenoside Rb1 upregulates BDNF expression in hippocampi of rats subjected to acute stress [29,30]. Rg1 also regulates BDNF expression and attenuates depression-like behaviors induced by chronic stress [26,31], and Rg3 and Rg5 similarly exhibit antidepressant-like effects in mice by activating hippocampal BDNF signaling [32,33]. Based on the neuroprotective properties of multiple ginsenosides and taurine via regulation of BDNF expression, we examined the abilities of ginsenoside Rf (g-Rf) and taurine, alone and in combination, to induce BDNF expression in human SH-SY5Y neuroblastoma cells.

## 2. Results

### 2.1. g-Rf and Taurine Induce BDNF Expression

The viability of SH-SY5Y cells treated with various doses of g-Rf and taurine was determined by MTT assays. No cytotoxic effects were observed when the cells were exposed for 24 h to increasing concentrations of these compounds alone (Figure 1A,B) or in combination (10 μM g-Rf and 25 μM taurine) (Figure 1C).

Exposure of SH-SY5Y cells to g-Rf increased the levels of BDNF mRNA and protein in a time- and concentration-dependent manner. Treatment with 10 μM g-Rf significantly increased BDNF mRNA levels within 2 h (*p* < 0.05) and to a maximum of approximately 20-fold at 8 h (*p* < 0.01), before declining slightly (Figure 2A). Significant increases were obtained after treatment for 8 h with doses of at least 2.5 μM g-Rf, with maximal induction at 10 μM (*p* < 0.01) (Figure 2B); thus, 10 μM g-Rf was chosen for subsequent experiments. The increases in BDNF mRNA resulted in corresponding time-dependent (Figure 2C) and dose-dependent (Figure 2D) increases in protein levels.

Similar time- and dose-dependent increases in BDNF mRNA and protein were observed in SH-SY5Y cells treated with taurine, with maximal effects at 4–8 h and at a concentration of 25 μM (Figure 3). However, the magnitudes of the effects were smaller than that observed with g-Rf.

### 2.2. Ginsenosides and Taurine Synergistically Induce BDNF Expression

We next assessed BDNF induction when SH-SY5Y cells were treated with taurine in combination with ginsenosides. BDNF mRNA levels were significantly higher in cells treated with ginsenoside Re, Rf, Rg1, or Rh1 (all 10 μM) in combination with 25 μM taurine than in cells treated with taurine alone (*p* < 0.05) (Figure 4A). The largest increase was observed when cells were exposed to taurine and g-Rf, which increased the expression of BDNF mRNA by 1.5-fold compared with that by g-Rf alone (*p* < 0.01) (Figure 4B). Accordingly, taurine and g-Rf synergistically increased the level of BDNF protein (Figure 4C).

### 2.3. g-Rf and Taurine Upregulate BDNF Expression via MAP Kinases

To identify the mechanism by which g-Rf and taurine induced BDNF expression, we assessed the activation of three mitogen-activated protein (MAP) kinases: p38, ERK, and c-Jun N-terminal kinase (JNK). The phosphorylation of p38 and ERK, but not JNK, was increased in SH-SY5Y cells treated with g-Rf and taurine, with maximal levels observed after 30 min and effects sustained for at least 240 min (Figure 5A).

To determine if the activation of these kinases was critical for the induction of BDNF, cells were treated with g-Rf and taurine in the presence of inhibitors of p38, ERK, or JNK. As shown in Figure 5B, the increase in BDNF protein induced by g-Rf and taurine was suppressed by 10 μM SB203580 and PD98059, which block p38- and ERK-mediated signaling, respectively; inhibition of the JNK pathway with 10 μM SP600125 did not prevent BDNF induction. The viability of cells treated with SB203580 (10 μM), SP600125 (10 μM), or PD98059 (10 μM) was >90% as determined by the MTT assay (data not shown), demonstrating that the effect on protein induction was not a result of toxicity.

### 2.4. g-Rf and Taurine Protect against Corticosterone (CORT)-Induced Cellular Damage

Corticosterone (CORT) can induce neuronal damage [34]. As BDNF is neuroprotective [20,21,22], we examined whether g-Rf and taurine, which induce BDNF expression, are protective against CORT-induced cellular damage. The release of lactate dehydrogenase (LDH), an indicator of cellular damage, was significantly increased in SH-SY5Y cells exposed to 50 μM CORT (*p* < 0.01) (Figure 6A). However, this increase in LDH release was blocked when cells were pretreated for 6 h with g-Rf and taurine. An MTT assay further demonstrated the cytoprotective effect, as the viability of cells pretreated with g-Rf and taurine was significantly higher than those exposed only to CORT (*p* < 0.01) (Figure 6B). In addition, CORT-triggered morphological changes in neurites were also reversed in the presence of g-Rf and taurine (Figure 6C).

SH-SY5Y cells treated with 50 μM CORT showed a time-dependent decrease in BDNF mRNA expression (Figure 7A). However, this CORT-triggered decrease was drastically and significantly reversed in the presence of both g-Rf and taurine (*p* < 0.01) (Figure 7B,C). These results suggest that BDNF is a primary factor in the protective action of g-Rf and taurine against CORT-induced cellular damage.

## 3. Discussion

The results of the present study demonstrated that the ginseng-derived medicinal compound g-Rf and taurine synergistically increased the expression of BDNF in human neuroblastoma cells. This increase was mediated by ERK and p38 MAP kinase signaling and was sufficient to protect cells from CORT-induced damage. This is the first report, to our knowledge, of a direct effect of taurine with a ginsenoside on BDNF expression in SH-SY5Y cells.

BDNF plays a critical role in multiple pathological conditions, such as Alzheimer’s disease, Huntington’s disease, and psychiatric disorders [35]. The neuroprotective effects of BDNF are well-known [21,24,36,37], including against CORT-mediated damage in cellular and animal models of brain injury [38,39,40]. In addition, BDNF demonstrates neuroprotective effects in a CORT-induced model of depression in female mice [38]. Thus, the upregulation of BDNF by ginsenosides represents a mechanism by which they may exert their therapeutic effects. For example, the ginsenoside Rb1 upregulates hippocampal BDNF expression to relieve stress in rats subjected to stressful conditions [29,30], and Rg1, Rg3, and Rg5 activate BDNF signaling to elicit antidepressant effects [26,31,32,33]. Ginsenoside Rf also relieves depression-like behavior in rats with chronic constriction injury by alleviating neuropathic pain in the spinal cord [41]. Our work presented here shows that, similar to other ginsenosides, g-Rf upregulated BDNF expression and was able to reverse the CORT-mediated decreased of BDNF in SH-SY5Y cells, thereby mitigating cellular damage and the loss of viability.

Interestingly, we observed that taurine similarly increased, albeit to a lesser magnitude, BDNF expression in SH-SY5Y cells. This contrasts with a previous study that showed taurine decreased BDNF transcription in cells of the frontal cortices of rats chronically exposed to alcohol [42]. Moreover, taurine augmented g-Rf-induced increase in BDNF expression, and our experiments revealed that the mechanism for their effects involved the activation of ERK and p38 MAP kinase signaling. This is in line with the critical role of ERK survival signaling in the biological effects of taurine [43,44] and the involvement of p38 MAP kinase in taurine-mediated alleviation of myocardial oxidative stress in rats [45]. In addition, both ERK and p38 kinases are associated with immune activation by ginsenosides Rg3 and Rh2-B1 [46,47].

The ERK/p38-mediated upregulation of BDNF by taurine and g-Rf was involved in the attenuation of cell damage triggered by CORT, as the effects were eliminated by inhibitors specific to these kinases. g-Rf was originally shown to protect against amyloid-β-induced neurotoxicity in mice by regulating inflammatory responses [48]. Here, we demonstrated that g-Rf also exhibits beneficial effects in SH-SY5Y cells via its induction of BDNF expression. The protective effects of taurine and g-Rf against neuronal damage have been attributed to diverse biological actions [2,10]. Our results indicated that one mechanism could involve BDNF upregulation via ERK/p38 signaling, which mitigates CORT-triggered cellular damage. Thus, a combination of taurine and g-Rf could represent a potent therapeutic agent for neuronal damage.

## 4. Materials and Methods

### 4.1. Reagents

Ginsenosides Re, Rf, Rg1, and Rh1 were provided by Chengdu Biopurify Phytochemicals Ltd. (Chengdu, China). Thiazolyl blue tetrazolium bromide (MTT), taurine, and CORT were purchased from Sigma-Aldrich Co. (St. Louis, MO, USA). A monoclonal anti-BDNF antibody was purchased from Abcam (Cambridge, UK). Polyclonal antibodies specific for ERK, JNK, p38, and phospho-p38, as well as a monoclonal antibody specific for phospho-ERK were obtained from Cell Signaling Technology (Danvers, MA, USA). Monoclonal antibodies specific for α-tubulin and phospho-JNK were acquired from Santa Cruz Biotechnology (Dallas, TX, USA). PD98059, SB203580, and SP600125 were purchased from Calbiochem (La Jolla, CA, USA).

### 4.2. Cell Culture

Human neuroblastoma SH-SY5Y cells were supplied by the Korean Cell Line Bank (Seoul, Korea). Cells were maintained in a minimum essential medium containing Earle’s balanced salts (HyClone, Logan, UT, USA), 10% heat-inactivated fetal bovine serum (Life Technologies Corp., Carlsbad, CA, USA), 100 μg/mL streptomycin, and 100 U/mL penicillin at 37 °C in an atmosphere of 95% air and 5% CO_2_.

### 4.3. Western Blot Analysis

SH-SY5Y cells were exposed to each reagent for the designated times. After washing with ice-cold phosphate-buffered saline, the cells were lysed by pipetting in PRO-PREP protein extraction solution (iNtRON Biotechnology, Seongnam, Korea). The protein concentration was determined by the Bradford method using bovine serum albumin as a standard. An aliquot (15–20 μg protein per lane) of whole-cell lysate was fractionated by sodium dodecyl sulfate-polyacrylamide gel electrophoresis and then transferred to immobilon-P polyvinylidene difluoride membranes (Merck, Darmstadt, Germany). After brief washing with distilled water, membranes were blocked in a 5% skim milk solution prepared using Tris-buffered saline containing 0.1% Tween-20 (TBS-T). The membranes were then reacted with primary antibodies dissolved in TBS-T solution at 4 °C. After briefly washing with TBS-T, a peroxidase-conjugated secondary antibody was added to the solution, and the membranes were incubated for 1 h at ambient temperature. After thorough washing with TBS-T, the immunoreactive proteins were visualized using WesternBright ECL (Advansta Inc., Menlo Park, CA, USA).

### 4.4. Real-Time Polymerase Chain Reaction (PCR) Analysis

TRIzol (Invitrogen, Carlsbad, CA, USA) was used to extract total RNA from cells treated with the various reagents for designated times. The RNA was reverse-transcribed into cDNA using a TOPscript RT DryMIX kit (Enzynomics, Daejeon, Korea) and amplified in a 10 μL reaction mixture containing 1× PCR master mix (Solgent, Daejeon, Korea) and 0.5 μM primers. The conditions of real-time PCR using a LightCycler 96 instrument (Roche Diagnostics, Basel, Switzerland) were as follows: one cycle of initial denaturation for 15 min at 95 °C, 45 cycles of 15 s at 95 °C, 15 s at 55 °C, and 40 s at 72 °C. The primer sequences were as follows: BDNF, 5′-TGCAGGGGCATAGACAAAAGG-3′ and 5′-CTTATGAATCGCCAGCCAATTCTC-3′; and RPS18, 5′-TGCGAGTACTCAACACCAAC-3′ and 5′-GTCTGCTTTCCTCAACACCA-3′.

### 4.5. Cytotoxicity Assays

For the LDH release assay, a CytoTox 96 nonradioactive cytotoxicity assay kit (Promega, Madison, WI, USA) was utilized to determine the amount of LDH released into the culture medium of cells treated with reagents. A Multiskan GO microplate spectrophotometer (Thermo Scientific, Waltham, MA, USA) was used to measure the absorbance at 490 nm. For the MTT assay, after the cells were treated, they were incubated in a culture medium containing MTT solution (final concentration, 0.1 mg/mL) for 2 h. After the medium was removed, the formazan crystals were dissolved in acidified isopropanol solution and the absorbance was measured at 570 nm on the Multiskan GO microplate spectrophotometer.

### 4.6. Statistical Analysis

Data are expressed as mean ± standard error (SE). Statistical significance was determined by an unpaired *t* test with Welch’s correction using SigmaPlot 10 software (Systat, Chicago, IL, USA). A *p*-value of <0.05 was considered to be statistically significant.

## Figures and Tables

**Figure 1 molecules-25-02819-f001:**
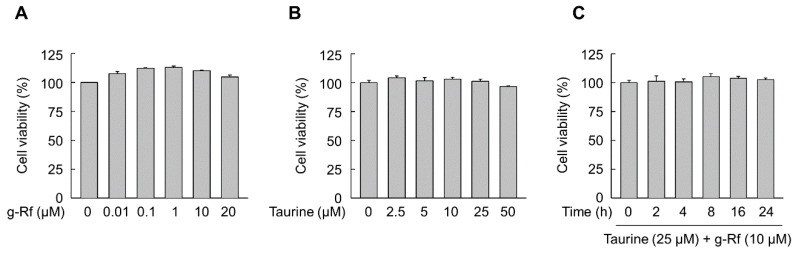
Viability of SH-SY5Y cells via MTT assays. Cells were incubated for 24 h with various concentrations of ginsenoside Rf (g-Rf) (**A**) or taurine (**B)**. (**C**) Viability over time was also assessed after exposure to a combination of g-Rf and taurine. Data are presented as mean ± SE.

**Figure 2 molecules-25-02819-f002:**
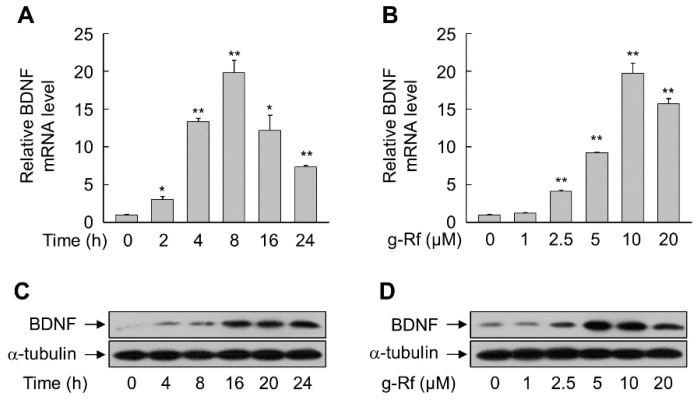
Induction of brain-derived neurotrophic factor (BDNF) in SH-SY5Y cells by ginsenoside Rf (g-Rf). (**A**,**C**) Cells were incubated with 10 μM g-Rf for the indicated time periods. (**B**,**D**) Cells were incubated with the indicated concentrations of g-Rf for 8 h (B) or 24 h (D). Total RNA and whole-cell lysates were prepared and subjected to real-time PCR (**A**,**B**) and immunoblot analysis (**C**,**D**). Data are presented as mean ± SE (*n* = 3 or 4). **p* < 0.05, ***p* < 0.01 compared with untreated group.

**Figure 3 molecules-25-02819-f003:**
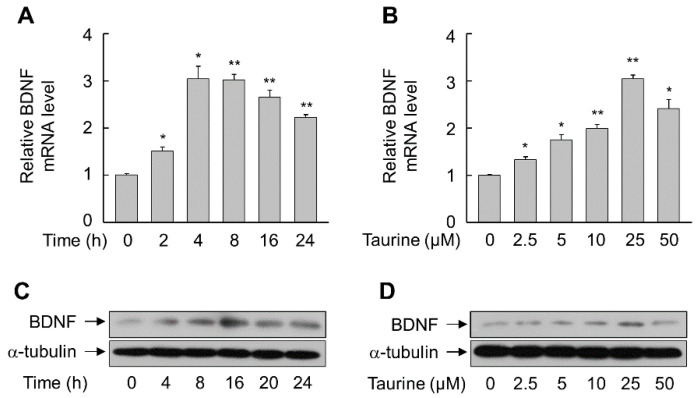
Induction of BDNF in SH-SY5Y cells by taurine. (**A**,**C**) Cells were incubated with 25 μM taurine for the indicated time periods. (**B**,**D**) Cells were incubated with the indicated concentrations of taurine for 8 h (B) or 24 h (D). Total RNA and whole-cell lysates were prepared and subjected to real-time PCR (**A**,**B**) and immunoblot analysis (**C**,**D**). Data are presented as mean ± SE (*n* = 3 or 4). **p* < 0.05, ***p* < 0.01 compared with untreated group.

**Figure 4 molecules-25-02819-f004:**
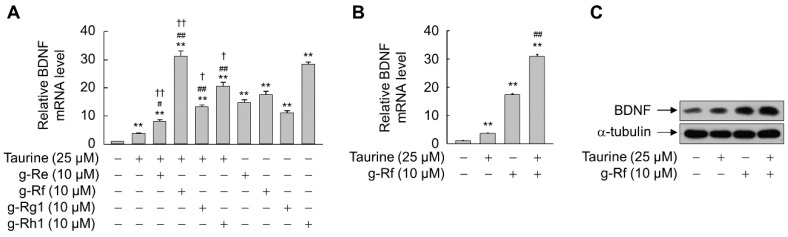
Synergistic effects of ginsenosides with taurine in upregulating BDNF in SH-SY5Y cells. (**A**) Cells were incubated with individual ginsenosides in the presence or absence of taurine for 8 h. (**B**,**C**) Cells were incubated with taurine and/or ginsenoside Rf (g-Rf) for 8 h (**B**) or 24 h (**C**). Total RNA and whole-cell lysates were prepared and subjected to real-time PCR (**A**,**B**) and immunoblot analysis (**C**). Data are presented as mean ± SE (*n* = 3 or 4). ***p* < 0.01 compared with untreated group; ^#^*p* < 0.05, ^##^*p* < 0.01 compared with taurine-treated group; ^†^*p* < 0.05, ^††^*p* < 0.01 compared with individual ginsenoside-treated group.

**Figure 5 molecules-25-02819-f005:**
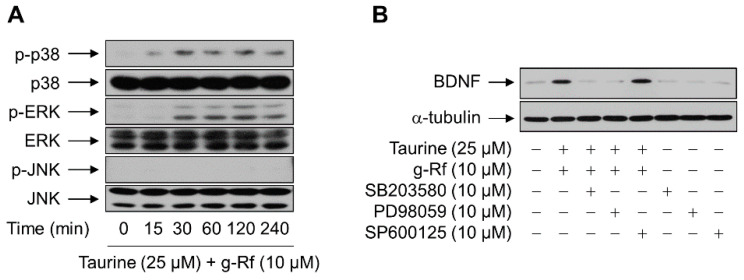
Roles of the mitogen-activated protein (MAP) kinase cascades in the upregulation of BDNF by taurine and ginsenoside Rf (g-Rf). (**A**) SH-SY5Y cells were treated with a combination of taurine and g-Rf for the indicated time periods, and whole-cell lysates were subjected to western blotting. (**B**) SH-SY5Y cells pretreated with vehicle (DMSO), SB203580, PD98059, or SP600125 for 1 h were incubated with or without a combination of taurine and g-Rf for 24 h, and whole-cell lysates were subjected to western blotting. α-Tubulin was adopted as a loading control.

**Figure 6 molecules-25-02819-f006:**
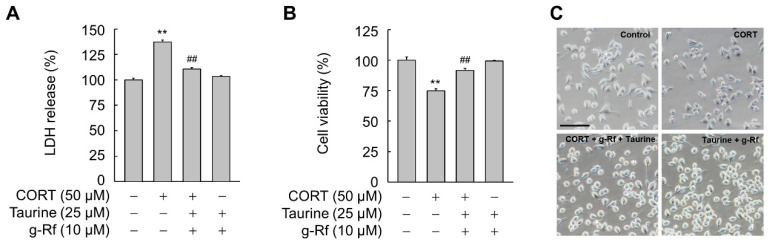
Effect of taurine and ginsenoside Rf (g-Rf) on corticosterone (CORT)-triggered cellular damage in SH-SY5Y cells. Cells were pretreated with the indicated concentrations of taurine and/or g-Rf for 6 h and then exposed to CORT for 24 h. Cellular damage was evaluated by lactate dehydrogenase (LDH) release assay (**A**) and MTT assay (**B**). (**C**) Cells pretreated with 25 μM taurine and 10 μM g-Rf for 8 h were incubated with or without 50 μM CORT for 24 h. The cells were photographed using a bright field optical microscope. Bar, 100 μm. Data are presented as mean ± SE (*n* = 3 or 4). ***p* < 0.01 compared with untreated group; ^##^*p* < 0.01 compared with CORT-treated group.

**Figure 7 molecules-25-02819-f007:**
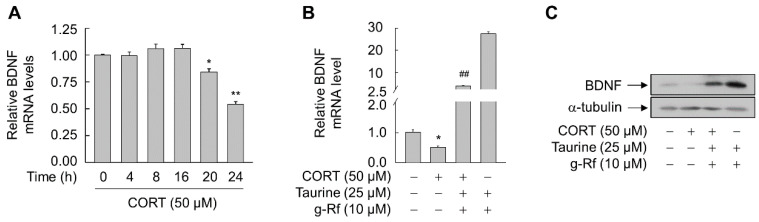
Effect of combined taurine and ginsenoside Rf (g-Rf) on corticosterone (CORT)-triggered expression of BDNF in SH-SY5Y cells. (**A**) Cells were treated with CORT for the indicated time periods, and extracted RNA was analyzed by real-time PCR to determine BDNF mRNA levels. Cells were pretreated with combined taurine and g-Rf for 6 h and then exposed to CORT for 24 h, and BDNF mRNA (**B**) and protein (**C**) levels were analyzed by real-time PCR and immunoblotting, respectively. Data are presented as mean ± SE (*n* = 3 or 4).**p* < 0.05, ***p* < 0.01 compared with untreated group; ^##^*p* < 0.01 compared with CORT-treated group.

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
