# Peer review of "Taurine and Ginsenoside Rf Induce BDNF Expression in SH-SY5Y Cells: A Potential Role of BDNF in Corticosterone-Triggered Cellular Damage"

_molecules, 2020, doi:10.3390/molecules25122819_

Round 1

Reviewer 1 Report

The authors report on the combined use of taurine and ginsenoside Rf to increase the expression of brain-derived neurotrophic factor (BDNF) in SH-SY5Y human neuroblastoma cells, which they show to be dose- and time-dependent.  They further address the underlying mechanisms and use appropriate inhibitors to demonstrate that ERK and p38 MAP kinase signaling are involved. Moreover, they provide evidence that in SH-SY5Y cells the combination of taurine and ginsenoside Rf has a protective effect against corticosterone (CORT)-induced BDNF suppression. While limited to a specific cell line, the results suggest that the potential of a combination of the two agents as a therapeutic option against neuronal damage merits further investigation.

The paper is well written and easy to read, and the experimental work was aptly done and interpreted. I have only a few minor issues:

1) On Fig. 2A it seems somewhat odd that the statistical difference of the relative BDNF mRNA level from the control is indicated as more powerful at 24 h (P<0.01) than at 16 h (P<0.05). The plot suggests otherwise. Also, can the authors suggest why there seems to be a decline after 8 h and ginsenoside Rf concentrations higher than 10 uM? And is this decline statistically significant?

2)  On Fig 4, A and B, the authors show statistically significant differences between relative BDNF mRNA levels in cells treated with combinations of ginsenosides and taurine compared with the taurine-treated group. However, to claim synergy, they must also demonstrate a statistically significant increase in the combination group(s) compared to ginsenoside alone. This is not clear from the information given in Fig. 4.

3) On the last paragraph of the discussion, the authors state that ‘The ERK/p38-mediated upregulation of BDNF by taurine and g-RF was a key event to attenuate the neuronal damage triggered by CORT…’

Perhaps they should refrain from using the terms ‘key event’. There may be other mechanisms involved that have yet to be identified.

Reviewer 2 Report

In this study the authors demonstrated that the treatment of SH-SY5Y neuroblastoma cells with taurine and ginsenoside Rf enhances BDNF expression and protects cells against corticosterone-induced damage throughout the activation of ERK and p38 MAP kinase signaling. The authors conclude that the combined treatment with both taurine and ginsenoside Rf exerts neuroprotective effects reducing neuronal damage triggered by corticosterone.

The study is well written, but the undifferentiated SH-SY5Y cells used for this study are cancer cells and do not represent an appropriate experimental model to evaluate neuroprotective effects. To this aim the authors need to perform all the experiments on SH-SY5Y differentiated cells or primary neuronal cells.

Reviewer 3 Report

The authors investigated the synergistic protective effects of taurine and ginsenoside Rf in corticosterone-treated human neuronal SH-SY5Y cells. Moreover, they studied its molecular mechanism. The BDNF expression was increased by taurine or ginsenoside Rf through the MAPK pathway. Moreover, taurine and ginsenoside Rf suppressed corticosterone-mediated cell damage. The results are sound. The manuscript was well-organized. However, the control study was missing. There are concerns that should be improved.

  1. The pictures of the cells treated with corticosterone together with taurine or ginsenoside Rf should be shown. The morphologies of these cells are important information.
  2. Combination of taurine or ginsenoside Rf synergistically activates the PAPK pathway. Taurine or ginsenoside Rf alone could activates the MAPK pathway, when each of them enhanced BDNF expression in Fig. 4B, C?
  3. In Fig. 5B, internal control should be also shown.
  4. 6B, the values of vertical axis are incorrect. If “%”, “100”, but not “1” is correct.
  5. What amount of proteins were loaded in each lane in Western blot analysis?
  6. The method for measurement of protein concentration should be indicated.

Round 2

Reviewer 2 Report

Although several previous studies used undifferentiated SH-SY5Y cells, it is well known that undifferentiated SH-SY5Y cells have an high proliferation rate and share only few properties with neurons, as also evident by the morphological features shown in figure 6C.

Therefore, the present study, aimed at elucidating the cell response (not neuronal cell response) to taurine and ginsenoside Rf is interesting.

However, the authors declare that in the present study they tried to elucidate the roles, but not neuroprotective roles, of taurine and ginsenoside Rf, while in the new version of the manuscript the effects of the g-Rf and taurine are still described as neuroprotective, both in the results and discussion sections. The term neuroprotection along the text is improperly used and confounding and must be changed.

For example:

-Line 151: the term neuroprotective in the sentence “BDNF is a primary factor in the neuroprotective action of g-Rf and taurine” must be changed in protective action.

-Line 176: “This is the first report, to our knowledge, of a direct effect of taurine with a ginsenoside on neuronal BDNF expression”, no neuronal cells have been used in this study, therefore the term neuronal must be changed with BDNF expression in SH-SY5Y cells.

Line 200: “attenuation of the neuronal damage triggered by CORT” , change in cell damage.

Line 202: “Here, we demonstrate that g-Rf is also neuroprotective via its induction of BDNF expression”, rephrase the sentence without referring to neuroprotective action.

Line 204: “We demonstrate here that one mechanism involves BDNF upregulation via ERK/p38 signaling”, this sentence must be rephrased “Our results indicate that one mechanism could involve BDNF upregulation via ERK/p38 signaling”.

Line 206: “a combination of taurine and g-Rf represents a potent therapeutic agent for neuronal damage”, change in “could represent”.

Author Response

As your indication, we have made corrections to avoid the term “neuroprotection” in the manuscript as follows:

① page 5, line 151 (underlined in RED), “BDNF is a primary factor in the protective action of g-Rf and taurine”.

② page 6, line 176-177 (underlined in RED), “This is the first report, to our knowledge, of a direct effect of taurine with a ginsenoside on BDNF expression in SH-SY5Y cells”.

③ page 6, line 199-200 (underlined in RED), “The ERK/p38-mediated upregulation of BDNF by taurine and g-Rf is involved in the attenuation of the cell damage triggered by CORT”.

④ page 6, line 202-203 (underlined in RED), “Here, we demonstrate that g-Rf also exhibits beneficial effects in SH-SY5Y cells via its induction of BDNF expression”.

⑤ page 6, line 205-206 (underlined in RED), “Our results indicate that one mechanism could involve BDNF upregulation via ERK/p38 signaling”.

⑥ page 6, line 206-207 (underlined in RED), “Thus, a combination of taurine and g-Rf could represent a potent therapeutic agent for neuronal damage”.

Reviewer 3 Report

All my concerns were met. The manuscript was improved.

Author Response

Thanks to your constructive comments.